# Multipurpose System for Cryogenic Energy Storage and Tri-Generation in a Food Factory: A Case Study of Producing Frozen French Fries



**Dimityr Popov** [1,*] , **Stepan Akterian** [2] , **Kostadin Fikiin** [1] **and Borislav Stankov** [1]

1  Faculty of Energy Engineering, Technical University of Sofia, 8 Sv. Kliment Ohridski Blvd.,
   1756 Sofia, Bulgaria; k.fikiin@tu-sofia.bg (K.F.); bstankov@tu-sofia.bg (B.S.)
2  Technological Faculty, University of Food Technologies, 26 Maritza Blvd., 4002 Plovdiv, Bulgaria;
   dr.akterian@gmail.com
*  Correspondence: dpopov@tu-sofia.bg

**Abstract:** This contribution elaborates on a futuristic hybrid concept for the multifunctional employment of a liquid air energy storage (LAES) system for combined heat, cold and power production (tri-generation) in a food factory, thereby providing a substantial part of the energy demand for various unit operations and enhancing the round-trip efficiency (RTE) of LAES. A processing line for frozen French fries, with relatively high heating and refrigeration demands, is used as a case study. The total useful energy output per charge/discharge cycle is 61,677 kWh (i.e., 38,295 kWh of electricity, 19,278 kWh of heating, and 4104 kWh of refrigeration). The estimated tri-generation RTE of the studied system reaches 55.63%, which appears to be 1.2 times higher than the RTE of a classical standalone LAES system with the same power input, considered as a baseline. In a broader context, such a performance enhancement by amalgamating food and energy technologies can make cryogenic energy storage a more viable grid balancing option capable of substantially increasing the share of renewables in the energy supply mix.

**Keywords:** refrigeration; food processing; cryogenics; LAES; renewable energy; polygeneration

## 1. Introduction

Cryogenic energy storage, and particularly liquid air energy storage (LAES), is receiving growing interest from the research community as one of the most promising technologies for large-scale energy storage in the power grid. However, real-world LAES applications are still quite limited, mainly because of the complex LAES plant layout, as well as because of the low ratio between the energy recovered from the LAES system during the discharge stage and the total energy input, known as the 'round-trip efficiency' (RTE). A detailed bibliography of numerous publications outlining recent trends in the field of LAES was presented by Borri et al. [1].

A typical LAES plant operates in three stages. In periods of excess power supply, which can come from intermittent renewable sources, electricity is used to liquefy air by means of compression and refrigeration (charge stage). The liquefied air is then stored in a low-pressure insulated tank (storage stage). In periods of high power demand, the liquid air is drawn from the tank, pumped and then heated by utilizing previously stored compression heat. The air is then expanded in a turbine connected to a generator, which supplies power to the grid (discharge stage).

Most often, the liquefaction technology for charging needs to be selected amongst the cycles of Linde–Hampson, Claude, Heylandt and Collins, or a cycle with two expanders. Shmeleva and Arkharov [2] compared different options and concluded that the Claude cycle is optimal for LAES purposes. Thus, the LAES configurations analysed hereafter in Sections 2.2 and 3.2 also resort to the Claude cycle.

Many recent studies in the area [3–8] clearly indicated that the discharging process cannot fully utilise the stored heat of compression in an efficient manner. According to She et al. [9], 20–45% of the compression heat is wasted, making this one of the main reasons for the poor RTE of a typical LAES system.

Many authors have studied different approaches for the utilization of the aforementioned excess heat. She et al. [7], Hamdy et al. [10] and Peng et al. [11] proposed a hybrid LAES system, involving an organic Rankine cycle (ORC) and a vapour-compression refrigeration cycle. The excess heat of compression is used as a heat source for the ORC, whereas the vapour-compression refrigeration cycle acts as a heat sink.

Tafone et al. [5] investigated nine options for internal and external use of the excess heat, including standalone electricity production and combined heat, cold and power production (tri-generation) which provides district cooling and heating. The best overall RTE of almost 56% is achieved by a configuration that uses a significant portion of the stored heat from compression to drive an absorption chiller and an ORC. In this configuration, the ORC generates additional electricity, while the absorption chiller and the air turbine with a heat exchanger generate chilled water at 6 °C to be used for air conditioning. It is worth mentioning that the baseline LAES system in their study had an RTE of 48.22%. Zhang et al. [12] integrated a Kalina cycle into the LAES system to utilise surplus heat during discharge in order to generate extra electricity, thus enhancing the RTE by some 5%.

Although such hybridizations improve the RTE, they introduce additional complexity to the LAES plant, which already contains a large number of components (i.e., compressors, turbines, heat exchangers, pumps, storage tanks, electrical generators, valves, and pipes). As a result, both the capital and the operation and maintenance costs of the system increase, while the daily cycling and seasonal dependence of such a complex system can be rather challenging.

Gao et al. [13] proposed a LAES system with tri-generation capability, which can provide district cooling, district heating and electricity during periods of peak power demand. Their results show an RTE improvement of 8% to 13% in the tri-generation case, as compared with a standalone system. She et al. [7] investigated a hybrid LAES configuration for combined district cooling, district heating, and hot water and power supply in decentralised micro energy networks. Their study concluded that such a configuration can achieve an overall RTE between 52% and 76%. A LAES-assisted polygeneration system for micro-grids applications has also been proposed by Mazzoni et al. [14], Howe et al. [15] focused on a combined, building-scale LAES and expansion system, while Szablowski et al. [16] analysed exergetically an adiabatic LAES system using the Linde–Hampson liquefaction cycle.

Fikiin et al. [17] proposed the idea of improving LAES performance and viability by directly utilising the cold released at the discharge stage in a refrigerated warehouse, thereby reducing the power consumed by the warehouse refrigerating plant and enhancing the overall RTE of the LAES system. This technology paves the path for a considerable storage and subsequent use of energy from renewable energy sources (RES) along the food cold chain, thereby decarbonising the sector by enhancing the sustainability of both refrigeration equipment and power grids [17]. Murrant and Radcliffe [18], Foster et al. [19], and Damak et al. [20] also envisaged this approach as a beneficial option for future LAES implementation.

Alongside refrigerated warehouses, food factories have significant potential for improving the RTE of a LAES system via tri-generation. The reason for this is that the technologies employed in food factories are energy-intensive and require both heating and refrigeration, along with electricity, at different stages of the production process. Popov et al. [21] proposed several options for utilizing the heat recovered from a LAES system for various food processing technologies requiring steam and/or hot water.

The specific energy consumption for manufacturing food products varies from 7 MJ/kg to 226 MJ/kg [22]. Food processing modes are energy-intensive and commonly employ electrical power or fossil-fuel boilers to heat steam or hot water, as well as to ensure refrigeration supply. Approximately 57% of the fossil fuel consumption in the

food industry is spent to generate steam and hot water [23]. Hence, when powered by renewables, LAES systems are capable of decarbonising both power grids and production processes by simultaneously providing heat, cold and electricity. Whereas the concept of LAES-enabled tri-generation, which meets the energy demands of a food factory, was originally proposed by Fikiin et al. [17] and Popov et al. [21] in a mostly qualitative manner, this promising idea has not yet been pursued in more detail to explore quantitatively a particular technology for food processing and preservation.

Consequently, the presented study aims to fill in the aforementioned knowledge gap by developing for the first time an innovative concept of enhancing LAES performance through its embedding in an integral tri-generation system supplying heat, cold and power to a food factory.

## 2. Material and Methods

### 2.1. Overall Approach

The following objectives were targeted: (i) selection of a reference standalone LAES system as a baseline case—named hereafter scenario A; (ii) identifying a typical food production line with both high heating and high refrigeration demands as a case study; (iii) specifying the requirements for the heating and refrigerating media and evaluating the heating and cooling loads for each unit operation along the food production line; (iv) developing the configuration of a LAES-based tri-generation system capable of meeting the heating and refrigeration demands of the selected food production line—named hereafter scenario B; (v) evaluating and comparing the RTEs for the above-mentioned scenarios A and B.

Figure 1 illustrates the methodological approach employed in simulating operational scenarios and determining the key performance characteristics of the suggested hybrid system by means of well-established underlying principles [24–26] and specialised commercial software [27].

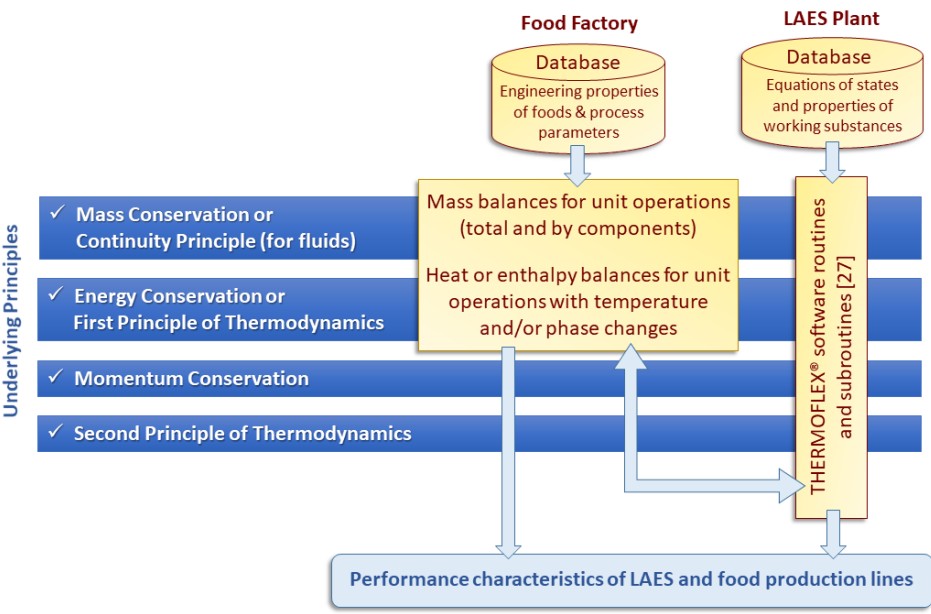

**Figure 1.** Overall methodology for simulating and quantifying the proposed hybrid system for the synergistic operation of an LAES plant and a food factory.

### 2.2. Standalone LAES System-Baseline Case

The thermodynamic performance of various LAES systems has been investigated by many researchers, particularly after 2010. Despite some design differences, almost all recently proposed LAES plants share a common architecture [9,28,29]. A typical configuration of a standalone LAES plant was retained (as depicted in Figure 2), which corre-

sponds structurally and parametrically to the configuration proposed by Guizzi et al. [28] and investigated in detail by Sciacovelli et al. [29]. This configuration represents the baseline case for the present study (scenario A).

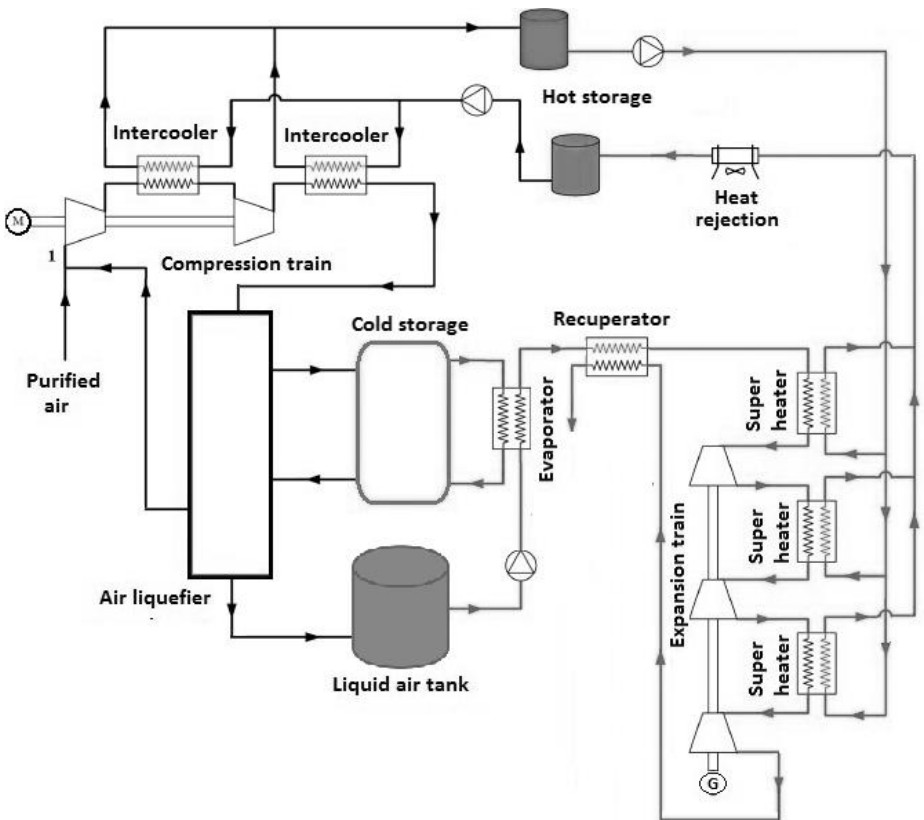

**Figure 2.** Typical configuration of a standalone LAES plant—scenario A.

During the charge stage, air is compressed in a two- or three-step process with intermediate cooling. Released compression heat is recovered by thermal oil, which is stored at a relatively high temperature in a hot storage section with a two-tank arrangement. Compressed air is afterwards cooled by the returning air from the air separator and by cold fluids supplied from the cold storage section, before flowing into a cryoturbine. The expansion in the turbine produces a vapour–liquid mixture that is collected and separated into a gas stream and a liquid stream in the air separator. The produced liquid air is then stored in an insulated tank at approximately 80 K and at atmospheric pressure.

During the discharge stage, the liquid air is pumped from a tank to high pressure and heated to a near-ambient temperature by the cold fluid circulating between the cold storage and the evaporator. This configuration permits some cold energy obtained during air regasification to be employed for cooling during the liquefaction stage, which is a highly energy-consuming process. Pumped air flows into a recuperator, where it is heated by the air leaving the expansion train. The expansion is usually divided into three stages with intermediate superheating, using thermal oil from the hot storage tank. The thermal oil is then cooled in a heat exchanger and returned to the hot storage section at an ambient temperate. This heat exchanger is essentially the only component of the plant where heat is rejected to the environment, since air is discharged from the recuperator at temperatures very close to the ambient one.

### 2.3. French Fries' Production as a Feasible Industrial Case for LAES Application

The food processing industry is a major consumer of heat, cold and power. Fossil fuel boilers supply steam or hot water for washing, blanching, concentration, pasteurization, cleaning-in-place, and other processes (Figure 3). Simultaneously, another major end

user of electricity in food processing is refrigeration, which is used for product cooling or freezing, refrigerated storage of raw materials and food products, and air conditioning of food facilities.

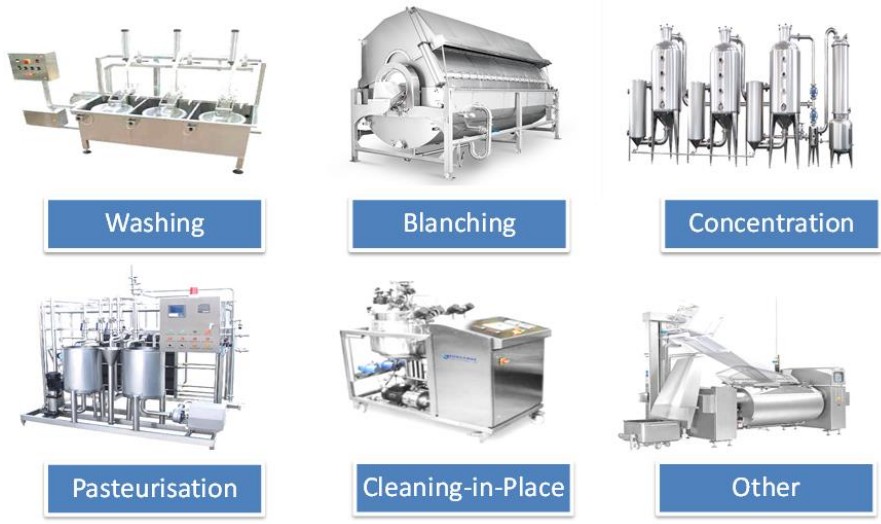

**Figure 3.** Examples of unit operations in a food factory with substantial heating demands [21]. Reproduced with the permission of the International Institute of Refrigeration (IIR).

A common line for the processing and storage of frozen French fries was chosen as an appropriate case study, because it encompasses a highly energy-intensive series of processes and associated equipment with both high heating and high refrigeration demands. The sequence of operations in such a line is shown in Figure 4.

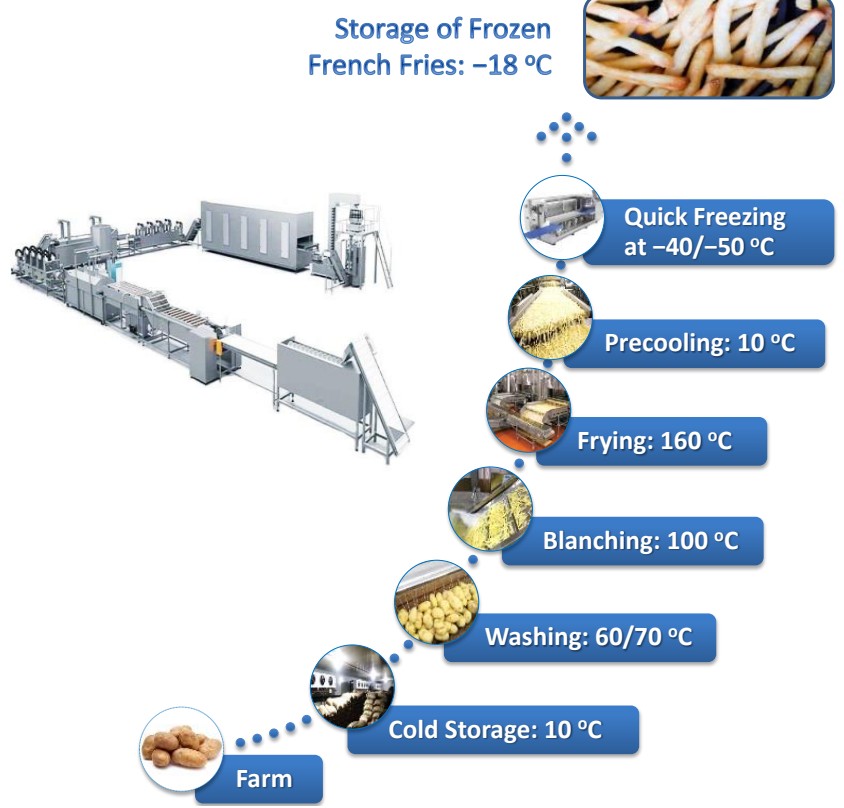

**Figure 4.** Main processes (unit operations) in a production line for frozen French fries involving substantial heating and refrigeration demands.

The capacity of the specified processing line is 2 tons of raw potatoes per hour, which is typical for medium-sized production lines of this kind. The incoming, intermediate and outgoing mass flows are shown in Table 1. The mass flow calculations assume 10% loss of mass during peeling and cutting, 10% oil uptake during frying and a 5% increase in mass due to packaging [30]. Changes in the relative moisture content of have you had the tickets he potatoes during processing (Table 1) are also taken into account.

**Table 1.** Production line for processing and storage of frozen French fries—mass flows and specific energy consumption for the considered unit operations.

| Operation | Cold Storage | Washing | Peeling and Cutting | Blanching | Frying | Pre-Cooling | Freezing | Packaging | Frozen Storage |
|---|---|---|---|---|---|---|---|---|---|
| Product flow | Raw potatoes | | Potato slices | | | French fries | | Frozen French fries | |
| Rel. moisture content, % | 75 | 75 | 75 | 75 | 75 | 48 | 45.5 | 45.5 | 45.5 |
| Mass flow, t/h | 2 | 2 | 1.80 | 1.80 | 1.80 | 1.10 | 1.10 | 1.12 | 1.12 |
| Mass flow, kg/s | 0.55 | 0.55 | 0.50 | 0.50 | 0.50 | 0.30 | 0.30 | 0.31 | 0.31 |
| Specific energy consumption, kJ/kg | 36.6 | 210.1 | N/A | 107.2 | 1795.8 | 463.7 | 209.4 | N/A | 12.9 |

Upon arrival at the production plant, the potatoes are usually cooled and remain for a while in a warehouse where a constant temperature of about 10 °C is maintained. At the beginning of the production process, the potatoes are washed in a hot water bath at 60–70 °C, where the water is indirectly heated by steam. After peeling and cutting, the potato slices are blanched in another steam-heated water bath at about 100 °C. This is followed by frying in a bath with sunflower oil at a temperature of 160–170 °C. The fried potato slices are then precooled with cold air to a temperature of 10 °C. Before packaging, the French fries are frozen to an end temperature of −18 °C in air blast freezing systems by using cold air at about −40 °C. The frozen fried potatoes are stored in a refrigerated warehouse at the same product temperature of −18 °C.

The target temperatures for each operation were specified according to Gould [31] and Singh and Kaur [32], as summarised in Table 2, along with the type and temperature of the heating or refrigerating mediums used for every process.

**Table 2.** Production line for processing and storage of frozen French fries—heating and refrigeration demands for the considered unit operations.

| Operation | Target Product Temperature, °C | Heating or Refrigerating Medium | Medium Temperature, °C | Heating Load, kW | Refrigeration Load, kW |
|---|---|---|---|---|---|
| Cold storage of raw potatoes | 10 | Cold air | 0–5 | N/A | 20.3 |
| Washing | 60–70 | Steam (1.2 bar) | 105 | 116.6 | N/A |
| Blanching | 100 | Steam (1.2 bar) | 105 | 53.6 | N/A |
| Frying | 160 | Steam (8 bar) | 170 | 897.9 | N/A |
| Precooling | 10 | Cold air | 0–5 | N/A | 139.1 |
| Freezing | −18 | Cold air | −40/−50 | N/A | 64.9 |
| Frozen storage of French fries | −18 | Cold air | −18/−20 | N/A | 4.0 |
| Total load | | | | 1068.1 | 228.3 |

## 3. Results and Discussion

### 3.1. Heating and Refrigeration Demands in a Typical French Fries' Production Line

The heating demands for washing and blanching and the refrigeration demand for precooling were evaluated by considering the mass flow of semi-products processed, their temperature changes and specific heat capacities. The latter were calculated according to Fikiin [33] and depending on the moistures and temperatures of the semi-product processed (see Tables 1 and 2).

Each load was multiplied by a correction coefficient to take into consideration the heat or cold dissipation into the environment. This coefficient varies from 1.02 to 1.06 and its value is higher for operations with a higher temperature difference between the heating medium and the environment. Heating demand for frying was estimated from the semi-product temperature change and the partial evaporation of water from the potato slices. The freezing load was calculated via the equations for mass specific enthalpy [34] depending on the temperature and moisture content of the semi-products.

The above correction coefficient was analogously employed for the operations of frying and freezing. The refrigeration loads for storing raw potatoes and frozen French fries were estimated by evaluating the heat losses through the walls, ceilings and floors of the storage rooms [26]. The capacity of these rooms was estimated with an assumption for 16 h of storage.

The total heating and refrigeration loads per production line were estimated to be 1068.1 and 228.3 kW, respectively. The specific energy consumptions for each operation are detailed in Table 1. The highest specific heating demand of 1.8 MJ/kg is during the frying process, which is carried out at the highest temperature. The lowest specific energy demands of 13–37 kJ/kg are observed during refrigerated storage. The overall specific energy consumption is 4.2 MJ/kg, which was estimated by taking into account the total heating and refrigeration demand of 1296.4 kW and the final product mass flow of 0.31 kg/s.

### 3.2. LAES-Powered Tri-Generation System

This publication focusses on assessing the capability of a LAES plant to supply additionally cold air and steam for frozen French fries' production lines, as described in Sections 2.3 and 3.1, by employing tri-generation. For the purposes of comparison, the baseline LAES plant configuration described in Section 2.2 is taken as a reference (Figure 2).

The analysis and design of the proposed hybrid tri-generation LAES system is performed by building on the strength of the already optimized baseline configuration (Section 2.2). The cold air demand of the French fry production line appears to be the critical parameter, which is most difficult to meet and thus characterizes the tri-generation plant capacity and size required. The discharge subsystem is redesigned and simulated in detail in a tailored manner by means of THERMOFLEX® [27]. The LAES plant's performance characteristics are thus identified and adjusted to match the targeted production process parameters (specified in Tables 1 and 2).

The reference LAES plant—i.e., scenario A—operates in charging mode when electricity is abundant and inexpensive (e.g., in periods of oversupply from intermittent RES). It is assumed that the LAES plant charges during a low-demand period of 9 h. The stored energy is supplied back to the grid over a 9 h peak-demand period during the day. Under nominal conditions, the reference LAES plant provides a power output of 5823 kW and an overall power storage capacity of 52,407 kWh. The power input is 12,319 kW, or 110,871 kWh per charge cycle.

The compression process involves two intercooled stages. Heat released is stored in sensible form using diathermic oil, acting as both a heat transfer fluid and a thermal storage medium. The oil in the hot storage tank is maintained at 374.2 °C, while in the lower-temperature thermal storage tank, it is maintained at an ambient temperature of about 20 °C. The heat input in hot storage is 11,800 kW or 106,200 kWh for every charge cycle.

During the discharge stage, the expansion is accomplished through three stages of air turbines with parallel intermediate superheating, using thermal oil from the hot storage tank with a temperature of 373.8 °C. The residual heat is rejected to the environment through a heat exchanger, where the thermal oil temperature is reduced from 185.4 °C to 20 °C. As a result, 41,220 kWh of heat are wasted.

The LAES plant with tri-generation capability—i.e., scenario B—is presented in Figure 4. As in scenario A, the plant charges during a low-demand period of 9 h. Power is returned back to the grid over a 9 h peak-demand period during the day, along with cold air and steam supplied to the food factory. The anticipated food factory has two production lines, each having cooling and heating demands as specified in Table 2. Such operation of the LAES plant provides all the energy needed by the food factory if it works in one shift. If the factory operates in two shifts, the energy needs of the second shift should be met by the conventionally installed facilities, such as a vapour-compression refrigeration plant and a fossil-fuel-fired boiler.

While the charge section of the LAES plant in scenario B has the same configuration as in scenario A, the discharge section layout is rather different. As specified in Table 2, the processes of freezing and frozen storage of French fries require cold air at low temperatures, which is produced by means of a single-stage air turbine. As shown in Figures 5 and 6, prior to its expansion in the turbine, the airflow rate from the evaporator, amounting to 12.11 kg/s, is superheated with thermal oil at a temperature of 315.2 °C to ensure a turbine outlet temperature of −40 °C. The heat input in the superheater is approximately 4100 kW. As a result, the thermal oil temperature is reduced from 373.8 °C to 259.5 °C. The turbine drives an electric generator with a power output of 4255 kW. The specific work for air liquefaction is 0.27 kWh per kilogram of liquid, and the optimal charging pressure is 180 bar, while the optimal discharge pressure is 74 bar.

Before rejecting heat in the cooling tower, the oil is directed to a supplementary thermal fluid boiler (Figures 5 and 7). Heat recovered in the boiler is used for steam production. The boiler is a U-shaped tube heat exchanger, where the high-temperature oil flows through the tubes, thus heating water from the feedwater tank and converting it into saturated steam.

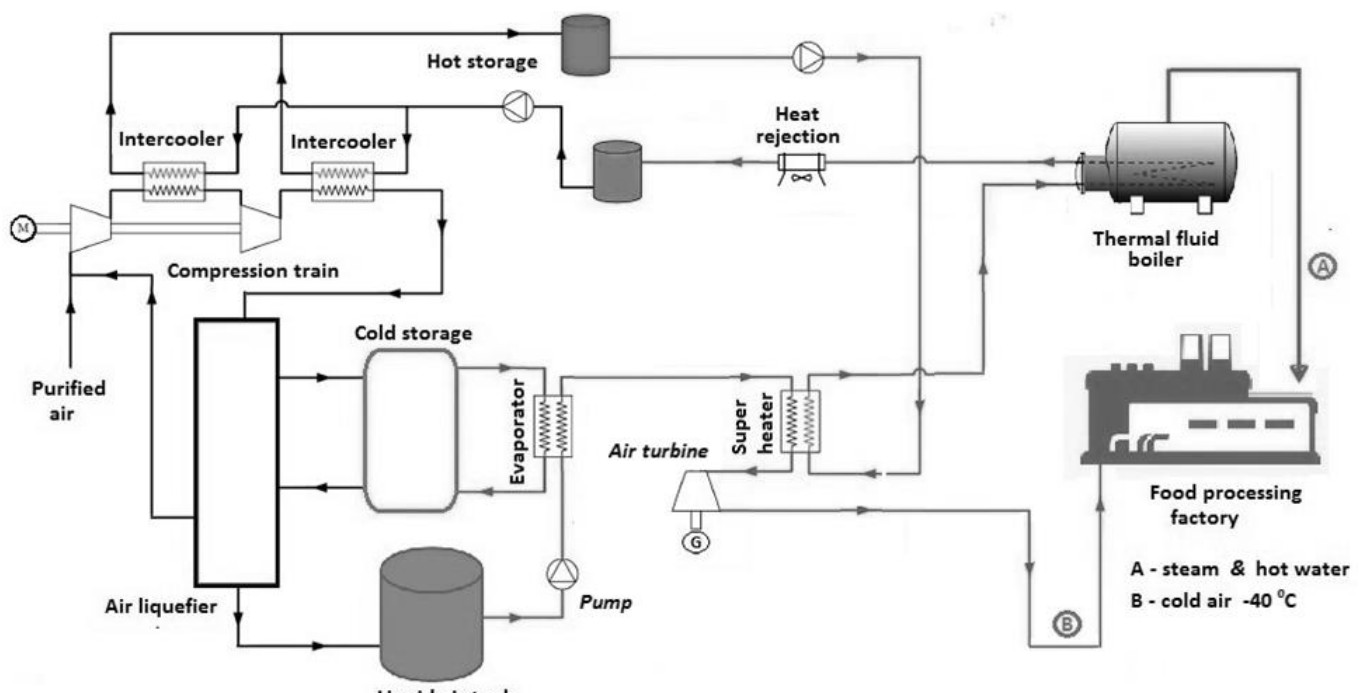

**Figure 5.** Layout of the LAES plant with tri-generation capability—scenario B.

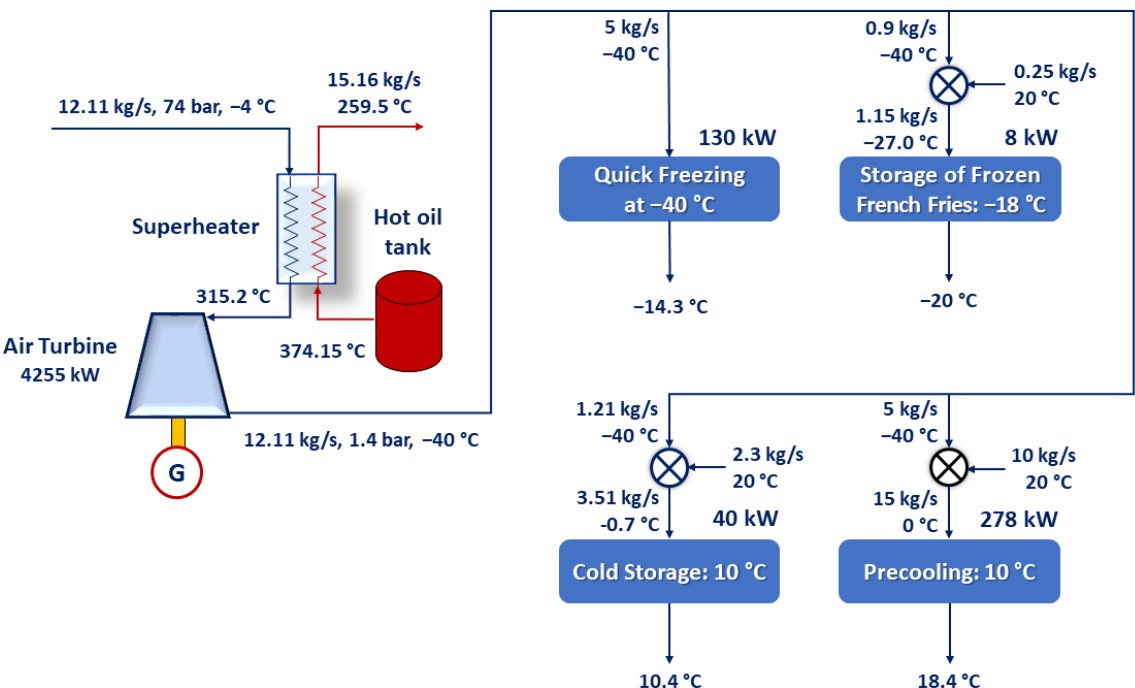

**Figure 6.** Possible arrangement of the cold air distribution among the consumers along the two identical French fry production lines.

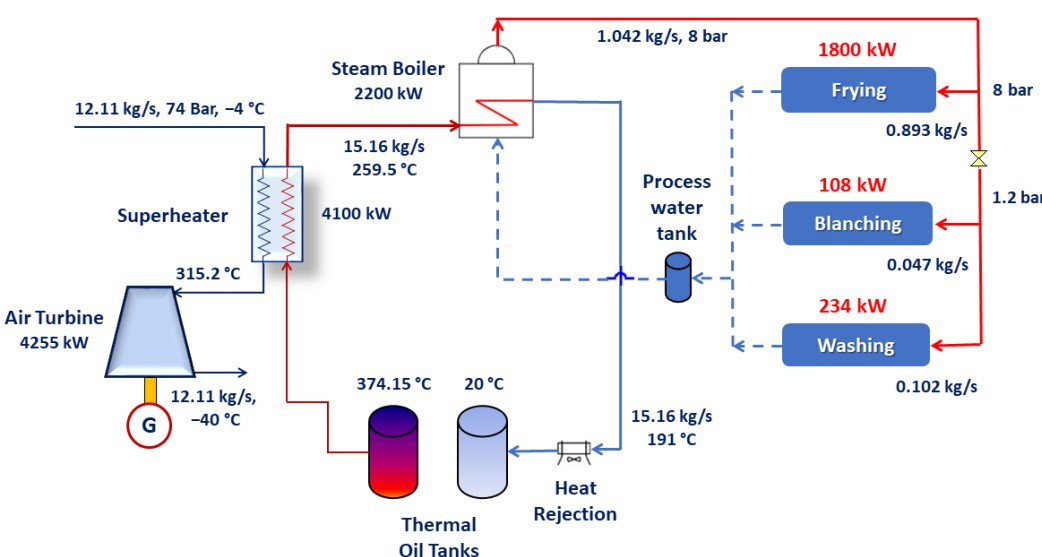

**Figure 7.** Steam generation and distribution among the consumers along the two studied French fry production lines.

The outlet cold air at −40 °C could be distributed along the French fry production lines by splitting the overall flow to four different portions connected in a parallel way (Figure 6). Freezing is accomplished by directly using a certain portion of the circulating air. The appropriate temperatures of the cold air for the rest of the consumers are achieved by mixing it with different quantities of fresh air at ambient temperature. The flowrates are adjusted to ensure that the refrigeration demand of each unit operation is properly met.

The heat recovery system is depicted in Figure 7. The cooled oil from the expansion turbine's superheater is directed to a steam boiler. Heat balance calculations were performed to determine the operating parameters of oil, steam, and condensate streams. The boiler generates saturated steam with a pressure of 8 bar. Most of the produced steam

is used to heat cooking oil for the frying process, which is effected via indirect-heated fryers. The remaining steam goes through a valve, where its pressure is reduced to 1.2 bar, and is then used for blanching and washing. The blanching steam heats the water baths in a continuous belt blanching machine, ensuring the uniformity of the processing temperature. The condensate from all consumers is returned to a process water storage tank.

The steam flowrate is determined to ensure that the heating demand of each unit operation is adequately met. The residual heat after the steam boiler is rejected to the environment by means of a heat exchanger, where the thermal oil temperature is reduced from 191 °C to 20 °C (Figure 7).

*3.3. Tri-Generation Versus Standalone Efficiency*

The RTE is defined as the ratio of the useful energy output (i.e., the utilized energy provided by the LAES plant during the discharge stage) to the power input (or the energy consumed by the LAES plant during the charge stage). For scenario A, the sole useful energy output is the power supplied to the grid during discharging. For scenario B, the useful energy output is the sum of the useful electrical output (power supplied to the grid), the useful heating output and the useful refrigeration output (heat and cold supplied to the food production line).

A comparison of the RTEs for the two scenarios is shown in Table 3. For the same power input into the LAES system, the RTE is boosted from 47.30% in scenario A (standalone power generation) to 55.63% in scenario B (tri-generation).

**Table 3.** Round-trip efficiencies of the LAES plant for scenarios A and B.

| Scenario | A | B |
|---|---|---|
| Power input, kWh | 110,871 | 110,871 |
| Power output, kWh | 52,407 | 38,295 |
| RTE of power generation only, % | 47.30 | 34.50 |
| Heating load, kW | N/A | 2142 |
| Useful heating output, kWh | N/A | 19,278 |
| Refrigeration load, kW | N/A | 456 |
| Useful refrigeration output, kWh | N/A | 4104 |
| Total useful energy output (heat, cold and power), kWh | N/A | 61,677 |
| RTE of tri-generation, % | N/A | 55.63 |

The RTE can potentially be increased further by a serial manner of refrigeration supply to the consumers of cold, unlike the parallel one illustrated in Figure 6. Although the serial organisation of cold energy distribution is generally more energy saving, it appears to be too hard to implement from the viewpoints of functionality and hygienic design. Another RTE-enhancing approach might be to recirculate the cold airflows outgoing from each unit operation (Figure 6) in a smart way. Nevertheless, such a fine and intelligent air-circulation control complicates the overall system, thereby involving a substantial increase in the capital costs.

## 4. Conclusions and Future Prospects

Cold-to-power systems are still rare in the energy sector, but reveal promising opportunities to make the energy system more flexible and integrative of renewables. In spite of their environmental friendliness, at today's technology level, standalone cryogenic energy storage systems (and specifically LAES) could still hardly compete with established energy storage principles as regards RTE. Thus, LAES needs to find niche areas in order to become a more viable and economical option by means of appropriate hybridisation and embedding into highly energy-intensive technologies and economic sectors.

Multipurpose renewable-powered LAES systems, employed for the poly-generation of heating, refrigeration and electricity, prevail in terms of functionality, flexibility and efficiency over the standalone systems producing electricity only. The present contribution proved this circumstance by exploring a case study of coupling LAES with a typical food factory producing frozen French fries. A tri-generation LAES configuration was proposed to produce power (for on-site or in-grid use) and to supply the factory with steam and cold air. By recovering the excess heat and cold, a tri-generation RTE of 55.63% could be reached, which exceeds 1.2 times the RTE of a typical standalone baseline system. A 9 h night charging cycle was assumed, which implies that wind power sources are a good candidate to feed renewable energy into the suggested LAES system.

The elaborated case study is the first of its kind, which addresses the potential synergy between LAES and food industry in a quantitative manner. Similar case studies might be found out in the food chains for meat, fish, dairy products, fruits, vegetables, etc. Drink production and more specifically breweries offer intriguing potential applications as well. Because of the vast diversity of specific unit operations, which feature various heat transfer, microbiological and biochemical scenarios, a uniform design approach towards all LAES-based engineering solutions would hardly be possible, while concrete processing, storage and distribution technologies need to be analysed individually on a case-by-case basis.

**Author Contributions:** Conceptualization: D.P. and K.F.; Methodology: D.P. and S.A.; Analysis: LAES plant D.P., K.F. and B.S.; food factory: S.A. and K.F.; Checking: K.F. and S.A.; Visualization: K.F. and B.S.; Writing, review and editing. D.P., S.A., K.F. and B.S. All authors have read and agreed to the published version of the manuscript.

**Funding:** The CryoHub Innovation Action received funding from the European Union's Horizon 2020 Research and Innovation Programme under Grant Agreement No. 691761.

**Acknowledgments:** The authors would like to thank all members of the CryoHub project consortium for the fruitful partnership and the useful discussions.

**Conflicts of Interest:** The authors declare no conflict of interest.

## Acronyms

| | |
|---|---|
| IIR | International Institute of Refrigeration |
| LAES | Liquid Air Energy Storage |
| ORC | Organic Rankine Cycle |
| RTE | Round-Trip Efficiency |
| RES | Renewable Energy Sources |

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
