# Peer review of "Multipurpose System for Cryogenic Energy Storage and Tri-Generation in a Food Factory: A Case Study of Producing Frozen French Fries"

_applsci, doi:10.3390/app11177882_

Round 1

Reviewer 1 Report

  • Many sentences are too long. Please, shorten them
  • The Nomenclature is missing
  • Introduction section: “All recent studies in the area clearly indicate that the discharging process cannot fully utilize the stored heat of compression in an efficient manner.”

What studies?

  • Introduction section: “As a result, 20 to 45% of the compression heat are wasted, making this one of the main reasons for the poor RTE of a typical LAES system.”

According to?

  • The main findings of Hamdy et al. (2019) and Peng et al. (2018) need to be presented in the Introduction section
  • In Section 1 the Authors need to clearly state what knowledge gap their work will fill COMPARED TO the current status on the investigated topic
  • The article needs to be subdivided into the following sections: Introduction, Methodology, Results and Discussion, Conclusions and Future work
  • “Cleaning-in-Place (CIP)” rather than “CIP (Cleaning-in-Place)” in Section 3
  • How can the results obtained be generalized/extended to other case studies?
  • Please, compare the proposed solution to the other solutions available in literature
  • The units of measurement are incorrectly formatted, e.g. kg*s^-1 rather than kg/s
  • Please, ass a suitable reference to THERMOFLEX in Section 5
  • The limitations and the necessary future developments of the work need to be summarized in Section 7

Reviewer 2 Report

The article presents an interesting concept of using the LAES for the needs of frozen French fries factory. The modeled system is to work in the trigeneration. Despite the fact that the idea itself is interesting, the article requires significant improvement in order to meet the requirements currently posed in the world of science.
Below I have described a number of corrections that should be introduced before publication:

  • Please extend the literature review to include articles from MDPI on LAES (may also be different from Applied Sciences - e.g. Energies).
  • Please include other liquefaction cycles in LAES systems in the literature review.
  • Please add a theoretical section describing the mathematical models of the LAES components you used. There is not a single equation in your article.
  • Please describe the mathematical models of the working factors used in your system.
  • Please list the liquefaction work and compare it with other literature.
  • Please put more information (than just efficiency) in the abstract about the modeled system, e.g. liquefaction work, system parameters.

Round 2

Reviewer 1 Report

The paper can be accepted for publication

Author Response

Thank you very much for the positive evaluation!

Reviewer 2 Report

The article has improved significantly. It's much better to read it now, although I still haven't found a section on calculations (only the methodology). There is no mathematical description of the energy balances carried out. My recommendation is as follows: the article needs some minor corrections before publication.

Author Response

We take the liberty to repeat that the methodology uses readily available IT instruments and trivial routines to simulate scenarios and obtain numerical results. As commercial software was used to generate most of the results, the authors do not pretend for any original merits in mathematical modelling. Artificial inclusion of equations would not add any value to the paper, while the focus would entirely be lost.

Anyway, we have additionally included Figure 1 to explain the methodology in much more details, thereby making things absolutely clear for the readership.